# Ionomic Differences between Susceptible and Resistant Olive Cultivars Infected by *Xylella fastidiosa* in the Outbreak Area of Salento, Italy

**DOI:** 10.3390/pathogens8040272

**Published:** 2019-11-28

**Authors:** Giusy D’Attoma, Massimiliano Morelli, Pasquale Saldarelli, Maria Saponari, Annalisa Giampetruzzi, Donato Boscia, Vito Nicola Savino, Leonardo De La Fuente, Paul A. Cobine

**Affiliations:** 1Department of Soil, Plant and Food Sciences, University of Bari Aldo Moro, 70126 Bari, Italy; giusy.dattoma@ipsp.cnr.it (G.D.); annalisa.giampetruzzi@uniba.it (A.G.); vitonicola.savino@uniba.it (V.N.S.); 2Italian National Research Council, Institute for Sustainable Plant Protection, 70126 Bari, Italy; massimiliano.morelli@ipsp.cnr.it (M.M.); pasquale.saldarelli@ipsp.cnr.it (P.S.); maria.saponari@ipsp.cnr.it (M.S.); donato.boscia@ipsp.cnr.it (D.B.); 3Department of Entomology and Plant Pathology, Auburn University, Auburn, AL 36849, USA; lzd0005@auburn.edu; 4Department of Biological Sciences, Auburn University, Auburn, AL 36849, USA

**Keywords:** calcium, ionome, leccino, manganese, Ogliarola salentina, olive, Olive quick decline syndrome, *Xylella fastidiosa*

## Abstract

Olive quick decline syndrome (OQDS) is a devastating disease of olive trees in the Salento region, Italy. This disease is caused by the bacterium *Xylella fastidiosa*, which is widespread in the outbreak area; however, the “Leccino” variety of olives has proven to be resistant with fewer symptoms and lower bacterial populations than the “Ogliarola salentina” variety. We completed an empirical study to determine the mineral and trace element contents (viz; ionome) of leaves from infected trees comparing the two varieties, to develop hypotheses related to the resistance of Leccino trees to *X. fastidiosa* infection. All samples from both cultivars tested were infected by *X. fastidiosa,* even if leaves were asymptomatic at the time of collection, due to the high disease pressure in the outbreak area and the long incubation period of this disease. Leaves were binned for the analysis by variety, field location, and infected symptomatic and infected asymptomatic status by visual inspection. The ionome of leaf samples was determined using inductively coupled plasma optical emission spectroscopy (ICP-OES) and compared with each other. These analyses showed that Leccino variety consistently contained higher manganese (Mn) levels compared with Ogliarola salentina, and these levels were higher in both infected asymptomatic and infected symptomatic leaves. Infected asymptomatic and infected symptomatic leaves within a host genotype also showed differences in the ionome, particularly a higher concentration of calcium (Ca) and Mn levels in the Leccino cultivar, and sodium (Na) in both varieties. We hypothesize that the ionome differences in the two varieties contribute to protection against disease caused by *X. fastidiosa* infection.

## 1. Introduction

Olive quick decline syndrome (OQDS) is a severe disease that was first reported in September 2013 in Salento, a region in the south of Apulia, Italy. The etiological agent of the disease is the bacterium *Xylella fastidiosa* [1,2]. *Xylella fastidiosa* is transmitted by xylem-feeding insects and causes well-known diseases such as Pierce’s disease (PD) of grapevine and citrus variegated chlorosis (CVC) of citrus. Additional host plants of economic and landscape relevance include almond, oleander, peach, coffee, oak, and blueberry [3]. The symptomatology, which varies depending on the crop, is represented by scorch and marginal necrosis of the leaves, chlorosis, and progressive desiccation of twigs and branches. Although the mechanism by which the bacterium causes disease is not completely understood, the most recognized hypothesis concerns the blockage of the plant xylem vessels that are used to transport water and mineral elements from the soil to the aerial parts. The development of the disease depends on the ability of the pathogen to move from the point of inoculation, proliferate within the xylem vessels, and colonize the whole plant [4]. The production of biofilm associated with bacterial multiplication, together with the production of gums and tyloses by the plant as a defense mechanism, could contribute to the obstruction in the vessels [5,6]. Different molecular pathways are induced by infection. Typical defense genes against pathogens, genes for the synthesis of hormones associated with biotic stress, genes known to be involved in the tolerance to water stress, as well as those involved in synthesis of hormones produced during abiotic stress, have all been detected as a response of *Vitis vinifera* and *Citrus reticulata* to *X. fastidiosa* infection [7,8,9]. Genome analysis of different *X. fastidiosa* strains revealed the absence of genes encoding Type III secretion system machinery, as well as the lack of apparent Type III secretion effectors, which are responsible for suppressing host plant defense responses [10]. Moreover, *X. fastidiosa* lives in xylem vessels, which are mainly composed of dead cells [11]. For this reason, key questions regarding plant–pathogen interaction are how the host senses the pathogen and what are the mechanisms triggered by this recognition.

While some species show severe disease, other species growing in the same environment show tolerance, resistance, or even immunity to the pathogen [12]. These hosts can harbor high populations of *X. fastidiosa*, but show few or no symptoms of the disease. One example is *Vitis rotundifolia* (or Muscadine grape), native to the southeastern region of North America, that has evolved resistance to the disease. In this region, Pierce’s disease is so severe that it prevents the commercial cultivation of many cultivars of *Vitis vinifera* [3]. Ruel and Walker identified the California native *Vitis girdiana* as resistant, which means it can support up to one hundred times higher bacterial concentrations without disease development [13]. Additionally, *Vitis arizonica*, also from an area of high infection pressure, appears to be very resistant to *X. fastidiosa* [13]. However, the mechanism of resistance is unknown in all these cases.

The OQDS is characterized by initial leaf scorch and scattered desiccation of small branches that over time get worse and extend to the whole canopy [14]. Nevertheless, greenhouse and field observations have revealed that plants of *Olea europaea* (olive) “Leccino”, although infected, appear asymptomatic or show very mild symptoms when compared with the highly susceptible and prevalent variety “Ogliarola salentina”, which is severely damaged and succumbs to infection. In particular, the canopy of Ogliarola salentina trees affected by the disease shows progressive and complete dieback, while infected Leccino trees show a better vegetative growth with less aggressive and more sporadic desiccation around the canopy [2,15]. Transcriptome profiling of the two olive cultivars in response to infection by *X. fastidiosa* subsp. *pauca* strain De Donno revealed that the pathogen triggers a differential response involving a re-modelling of cell wall proteins [16]. An up-regulation of genes encoding receptor-like kinases (RLK) and receptor-like proteins (RLP) was observed in Leccino, while genes related to drought stress were expressed in Ogliarola salentina [16]. These signaling proteins, whose orthologs are similarly up-regulated in CVC-resistant mandarin [9], are the first barrier that the plant produces against the pathogen. Therefore, it was assumed that Leccino variety is somehow able to respond to the pathogen. Moreover, trees of Leccino variety harbor a 100-fold lower bacterial population when compared with Ogliarola salentina [16], similar to resistant *Vitis* species and varieties of citrus in which multiplication and translocation of *X. fastidiosa* is inhibited [4,17,18,19].

Mineral and trace elements play a role in a myriad of cellular functions, acting as protein cofactors in many processes [20]. Therefore, mineral transport and balance is important for growth and development of plants and also microorganisms, and can be an important factor in disease control and progression [12,20,21,22]. In host–pathogen interactions, the competition to acquire these elements is a phenomenon known as “nutritional immunity” [23]. Analysis of the ionome, defined as the complete profile of the mineral nutrients and trace elements found in an organism [24], represents an approach to investigate the physiological state of the plant [25,26]. A number of experiments have demonstrated important roles of mineral elements for *X. fastidiosa* in vitro and during infection in planta. In previous studies, the analysis of the ionome of field-grown grapevines, blueberry, and pecan, and greenhouse-grown *Nicotiana tabacum* during *X. fastidiosa* infection have revealed significant changes between infected and uninfected plants, as well as between symptomatic and asymptomatic leaves, showing a complex interaction between different elements in the host [27,28,29]. Part of the complexity is related to the fact that virulence traits of *X. fastidiosa* are modulated by mineral elements. Bivalent cations, such as calcium (Ca) and magnesium (Mg), play a structural role in cell–cell and cell–surface attachment in the host, by acting as a bridge between the negative charges of xylem vessels and the negative charges of bacterial cells [30]. Additionally, in vitro assays proved that Ca promotes virulence determinants in *X. fastidiosa*, like biofilm formation, adhesion, and twitching motility [31,32]. In fact, *X. fastidiosa* cells from biofilm and planktonic states showed a different profile of mineral elements, with biofilm cells accumulating Ca, copper (Cu), potassium (K), Mg, and zinc (Zn), when compared with planktonic cells [33].

Here, we describe an empirical field survey, carried out in privately owned olive groves located in the *Xylella*-infected area in southern Italy, where we characterize the leaf ionome of two olive cultivars showing different disease responses to *X. fastidiosa* infection. Our working hypothesis was that the mineral profile of each cultivar plays a role in the success of the infection and/or in the development of symptoms. The very high levels of *Xylella* infections in the sampled fields make it unfeasible to obtain with absolute certainty uninfected control olives in these orchards. The study provides insights into some potential disease resistance mechanisms of the Leccino variety.

## 2. Results

### 2.1. Field Survey

Leaf samples from naturally infected olive trees were collected from two different olive groves located in the area declared *X. fastidiosa*-infected (Figure 1). All leaves from the two different cultivars of olives, Ogliarola salentina and Leccino, were naturally infected with *X. fastidiosa.* Ogliarola salentina showed severe symptoms, while symptoms on Leccino were less prominent and were often limited to initial discolorations that had not yet advanced to leaf scorching (Figure 2). Leaves designated as infected symptomatic showed initial desiccation, which consists of browning in color and “rolling” downward of the leaves. Our surveys showed no uninfected trees exist in these fields to serve as a control for infection status.

### 2.2. Population Size of Xylella fastidiosa within the Two Cultivars

*Xylella fastidiosa* qPCR detection confirmed that all plants analyzed were infected, despite the presence or not of symptoms by visual inspection (Figure 2B). The estimation of bacterial concentration showed significant differences among tissues of Ogliarola salentina and Leccino (Figure 3).

### 2.3. Determination of Leaf Ionome and Soil Parameters

The total concentrations of mineral elements for the leaves sampled are shown in Table 1. The elemental composition was compared with reference values for leaf content of olive trees [34]. The values for Ca, Mg, S, P, Na, and Zn were within these reference ranges. However, K and Fe were considered either low or close to the minimal range. Only K was in the range specified for potential deficiency (K < 4 g/kg), but this was true for all samples, independent of cultivar or symptomatology. Soils from Field 1 and Field 2 were analyzed to investigate their physicochemical parameters as a potential variable in differences in elemental composition. The region of Field 1 where the Leccino variety was grown, when compared with the Ogliarola salentina region, showed higher Ca (2230 vs. 1601 mg per kg) and lower Mn (26 vs. 38 mg per kg). While in Field 2, the soil from where the Leccino variety was grown compared with soil from the Ogliarola salentina region showed lower Ca (2130 vs. 2601 mg per kg) and higher Mn (56 vs. 26 mg per kg). The Ca concentration was lower than the reference of 3000 mg per kg, while the Mn concentration was higher than the reference of 5–20 mg per kg (Table 2).

### 2.4. Ionome of Leccino and Ogliarola Salentina

Principal component analysis suggests that Leccino and Ogliarola salentina have differences in their mineral and trace element content (Figure 4A,C and Figure 5A,C). The variance explained by each principal component is shown in Table 3, and the component loadings for PC1 and PC2 are shown in Table 4. The PCA component values were used for hierarchical cluster analysis and generation of a heat map (Figure 4B,D and Figure 5B,D). In general, the varieties were separated into independent clades with some overlap in ionome characteristics. We did not determine the variables contributing to this overlap/clustering together.

Specifically, comparing infected asymptomatic leaves in Field 1, Leccino had higher levels of K (23%), Mn (43%), Na (30%), and Zn (39%) and lower levels of Fe (−47%) and P (−15%) (*p* < 0.05 in all cases), compared with Ogliarola salentina (Table 5). In infected symptomatic leaves in Field 1, Leccino showed higher levels of Ca (21%), K (35%), Mn (41%), and lower levels of Fe (−37%) and P (−23%) (*p* < 0.05 in all cases), compared with Ogliarola salentina (Table 5). In Field 2, infected asymptomatic leaves of Leccino had higher levels of Fe (51%), Mn (50%), and lower levels of Ca (−40%), Mg (−34%), and Zn (−34%) (*p* < 0.05 in all cases) (Table 5), compared with Ogliarola salentina. The infected symptomatic leaves from Field 2 of Leccino had higher levels of Mn (100%) and Fe (45%) and lower levels of Na (−61%) and Zn (−45%) (*p* < 0.05 in all cases) (Table 5), compared with Ogliarola salentina.

### 2.5. Ionome According to Symptomatology

In Field 1, the comparison between infected symptomatic versus infected asymptomatic leaves of Ogliarola salentina showed almost no statistical difference in elemental composition, except for those in Na levels that were higher by 139% (*p* < 0.001) in infected symptomatic leaves (Table 6). In Field 2, both Na (449%, *p* < 0.001) and Zn (72%, *p* < 0.05) levels were significantly higher (Table 6). Ionomic profiles of infected symptomatic versus infected asymptomatic leaves from Leccino in Field 1 showed statistically significant higher levels of Ca (19%) and Na (79%) (*p* < 0.05 in all cases) (Table 6). In Field 2, Leccino showed higher levels of Ca (76%), Fe (38%), Mg (56%), Mn (81%), Na (220%), S (8%), and Zn (45%) (*p* < 0.05 in all cases) (Table 6).

The factorial univariate ANOVA analysis of the interaction between three individual elements, Ca, Mn, and Na, with the independent variables field, variety, and status (i.e., symptoms) shows that for Ca, there is a significant effect for field (F = 16.39; df = 1; *p*-value < 0.001), status (F = 12.25; df = 1; *p*-value = 0.001), and a significant interaction for field*variety (F = 11.66; df = 1; *p*-value = 0.001) and field*status (F = 7.81; df = 1; *p*-value = 0.006). In the case of Mn, we observed a significant effect for field (F = 20.57; df = 1; *p*-value < 0.001), variety (F = 54.80; df = 1; *p*-value < 0.001), status (F = 22.18; df = 1; *p*-value < 0.001), and a significant interaction for field*variety (F = 7.11; df = 1; *p*-value = 0.008), field*status (F = 20.28; df = 1; *p*-value < 0.001), variety*status (F = 4.52; df = 1; *p*-value = 0.034), and field*variety*status (F = 5.47; df = 1; *p*-value = 0.02). For Na, significant effects were observed for variety (F = 12.08; df = 1; *p*-value = 0.001), status (F = 63.70; df = 1; *p*-value < 0.001), and significant interaction for field*variety (F = 13.12; df = 1; *p*-value = 0.0018), field*status (F = 9.03; df = 1; *p*-value = 0.003), variety*status (F = 12.49; df = 1; *p*-value < 0.001), and field*variety*status (F = 5.56; df = 1; *p*-value = 0.019). These data support the idea that analyzing a control from a different uninfected field outside the outbreak area would not provide a useful comparison for this study, as field has an effect on two of the major elements that are different.

## 3. Discussion

*Xylella fastidiosa* is continuing to emerge as an important and devastating bacterial pathogen for many crops, and so far no cure has been identified. The current strategies for management are vector control to limit spread of infection and the use of cultivars that show resistance to the pathogen in the field. The latter requires replacing susceptible varieties in current fields or incorporating the tolerant genotypes into breeding programs. The field survey presented here is the first description of the leaf ionome of two olive cultivars that respond differently to *X. fastidiosa* infection. The quantification of the bacterial population in the two varieties grown in the field under high infection pressure confirms the resistance of Leccino to *X. fastidiosa* infection since the growth of the bacterium is restricted [35,36], and this leads to milder symptoms when compared with Ogliarola salentina. Considering that replanting of the varieties is not always a timely or feasible approach, understanding the mechanisms of resistance could represent a critical step towards a more rapid response to *X. fastidiosa* outbreaks.

Due to the fact that *X. fastidiosa*-free plants of both varieties were not available in the fields from the outbreak studied here, we were unable to assess the changes in ionome in the presence/absence of the pathogen. Recent studies have been constrained by this same variable; our review of the literature confirms that no single study conducted in recent years in the area of the Salento outbreak reports conclusive data showing sampling of uninfected and infected trees in the same field. Most studies take samples from different fields or from different areas [37,38,39,40,41]. Therefore, we focused on the hypothesis that differences in the ionome of cultivar in both infected symptomatic and infected asymptomatic leaves are informative to the resistance of the Leccino variety. We also note that differences in ionome studied here may be affected by different time of infection of the trees analyzed, although fields for this study are located in the initial focus of the outbreak, we assume they were infected at similar time points. An internal denominator was used to calculate the concentration of mineral elements for this analysis to account for at least four possible sources of error when using direct measurement of dry weight as an external denominator: (1) difficulty making exact measurement of small quantities of leaves; (2) some samples had an unknown substance that remained after acid digestion, the amount was stochastic and variable in leaves, therefore differently contributing to dry weight. (It should be noted that no additional mineral elements of interest could be extracted from these insoluble materials with extended and repeated digestions—data not shown); (3) dry weights were potentially affected by variable extraction of water given the waxy coating on leaves; and (4) potential losses during sample introduction into plasma based on the performance of the instrument.

To assess the importance of mineral content for a potential role in resistance of Leccino, our criteria were: (1) the difference should be statistically significant in Leccino relative to Ogliarola salentina in both fields, (2) the magnitude of the change should be in the same direction in both fields. Assessing levels by these criteria, the element that consistently changed was Mn, with a higher level in the Leccino relative to Ogliarola salentina. The other elements that are different in both fields are Fe and Zn; however, these elements are increased in one field and decreased in the other. The reason for this contrasting behavior between fields is unknown, but we speculate that it cannot be protective. The analysis of soil compared with leaf ionome suggests that the plants are tightly regulating the levels of these elements, as one field has higher available Mn, and the other has lower Mn levels, yet Leccino maintains higher concentrations of the element. Therefore, the only element that is associated with resistance and we can suggest is providing potential protection for Leccino is Mn. In addition, a recent study by Scortichini et al. [42], analyzing the soil micronutrient content of farms located in the Gallipoli area, showed a low content of this element in the soil of all the fields investigated, perhaps suggesting a possible reason for susceptibility of Ogliarola salentina in the region. Mn is an essential micronutrient for plant metabolism and development. About 35 different enzymes use Mn, including phenylalanine ammonia lyase (PAL), a key enzyme in the metabolism of polyphenolic compounds such as flavonoids, phenylpropanoids, and lignin [43]. Recent studies showed a significant increase in total lignin in *X. fastidiosa*-infected Leccino compared with the sensitive Ogliarola salentina [41]. On this basis, elevated Mn in Leccino could allow for enhanced lignin and flavonoids production to help protect the host. Simpson et al. [44] have suggested that the uptake of Fe and other transition metal ions such as Mn, by *X. fastidiosa*, cause the exhaustion of essential micronutrients in xylem, leading to symptoms typical of variegated chlorosis in citrus trees. Our data suggest the elevated level of Mn in Leccino could more effectively counterbalance any deficit triggered by the bacterial infection. Other examples of Mn-dependent enzymes are superoxide dismutase [45], oxalate oxidase [46], and the oxygen-evolving complex of photosystem II (PSII) [47,48,49]. Production of reactive oxygen species (ROS), termed “oxidative burst”, is a hallmark of successful recognition of infection and activation of plant defense [50], since it can directly act as antimicrobial agent during the plant defense response [51], however, it also can cause damage if not controlled. Therefore, increased Mn for superoxide dismutase may help to balance the benefit and cost of ROS production.

Our previous studies have demonstrated that ionome remodeling occurs before the first symptoms of disease in the *Nicotiana tabacum* model [52]. The outcome of the present investigation shows that comparing infected asymptomatic and infected symptomatic tissues, the most evident change in all plants is increased Na. This result is in agreement with the significant increase in Na detected in symptomatic leaves of Phony peach diseased trees, also caused by *X. fastidiosa* [27]. *Xylella fastidiosa*-associated diseases are referred to as leaf scorch diseases, based on the common symptom of necrotic tissue at the margin of pathogen-infected leaves. A long-standing hypothesis states that leaf scorch is a consequence of obstructed water flow and the production of tyloses and gums by the host, that attempts to hinder the progress of the infection [5]. Although symptoms of water stress induced by *X. fastidiosa* are quantitatively and qualitatively different from that of water stress induced by reduced water availability, it has been shown both that the pathogen induces water stress and that external water stress conditions enhance the development of the disease [7,53]. Under drought conditions, maintenance of cell turgor may occur by ionic balance and osmotic adjustment, which implies a net accumulation of solutes and ions, in order to maintain a favorable water potential gradient [54]. Thus, a specific physiological role of Na ion in an osmotic adjustment in infected symptomatic leaves could be proposed. A specific Na uptake is an integral part of *Atriplex halimus* response to water stress [55]. *A. halimus* is a xerohalophyte plant species able to cope with low external water potentials [56]. It has been reported that Na is involved in the maintenance of mesophyll chloroplast structure, mainly in relation to granal stacking [57,58]. Sodium may also be involved in the regeneration of phosphoenolpyruvate in mesophyll chloroplasts [59], because a Na gradient across the envelope could be an alternative energy source for the active transport of pyruvate [60]. An increase in Na uptake could then be the consequence of a stress-induced decrease in the efficiency of the Na/pyruvate cotransport system.

Studies on a tobacco model of infection and initial field studies in grapes, blueberries, and pecans showed that Ca is higher in samples infected by *X. fastidiosa*, compared with non-infected, and this increase is even greater in the presence of symptoms [28,52]. In fact, Ca-oxalate crystals have been detected in the xylem vessels of grapes and coffee plants after *X. fastidiosa* infection [61], and Ca accumulates in vessels obstructed by the bacterium [62]. Data presented here, comparing infected symptomatic and infected asymptomatic leaves, showed an increase in Ca content in infected symptomatic tissues in Leccino. We cannot determine whether the difference in Ca at the leaf level corresponds to an increase in the ion in the cytoplasm or in the apoplast. The plant acquires Ca mainly from the soil through the root system before it reaches the shoots through the xylem, as the potential for this cation is favorable to its uptake from the roots [63,64]. Ca is an essential nutrient for the plant and plays numerous roles and functions. It acts as a counter cation in the vacuole [65], as the divalent cation it is required for structural roles in the cell wall and membranes, and it interacts with different phytohormones for cell division and expansion [66]. In addition, it plays a key role as a second intracellular messenger, able to trigger several downstream signaling pathways. Increases in Ca levels in the cytosol is a known plant defense response to the attack of a pathogen [67]. Given the higher concentrations in Ca in the leaves, we would suggest that this is a response from the extracellular medium rather than export from the organelles. For the reasons previously mentioned for Na, the increased Ca concentrations could also be the result of a water-stress response, but no Ca increase has been detected in studies that have analyzed the plant’s specific responses to water stress [68]. Analyses of grapevine gene expression with *X. fastidiosa* infection show an increase in gene expression for Ca transporters in the infected plants, but not in response to water deficit alone [7]. Similarly, citrus plants infected with *X. fastidiosa* show upregulation of genes involved in Ca signaling [9]. In addition, Rapicavoli et al. [8] explored grapevine responses to early infection by a wild-type *X. fastidiosa* and *wzy* mutant, and RNAseq analysis showed an upregulation of genes related to Ca sensing and signaling, in the wild-type strain, that was absent in plants when infected with the *wzy* mutant, which is unable to cause disease [8]. Interestingly, Ca-related genes are regulated in the two olive cultivars under investigation here based on the dataset of Giampetruzzi et al. [16]. Initial examination of the deposited data shows overexpression of CDPK1 and genes encoding calcium-binding proteins (CML19 and CML29) in infected Ogliarola salentina but not infected Leccino, which may suggest that this host is more sensitive to the *X. fastidiosa*-induced Ca changes and this is activating a cascade that results in the development of disease symptoms.

This study of two varieties in the field extends our knowledge about the differential response to *X. fastidiosa* infection. Since this is a newly established pathosystem, we lack details about host–pathogen interaction and the mechanisms of resistance to the pathogen. The analysis of the ionome reveals differences between the varieties. Although we did not analyze uninfected trees as they are not present in the fields sampled, we did observe that the Leccino variety has higher Mn levels, and this element regulates numerous pathways that could modulate disease symptoms. It is also notable that additional factors such as age of trees, severity of symptoms, and specific agronomical practices in the individual fields could be affecting the ionome composition. To mitigate these factors, the analysis was restricted to trees from within the same field, thus presumably subjected to the same agronomic practices, and we did not compare with uninfected controls from different regions that would be subject to even more of these confounding factors. This information provides a framework for using the ionome to gain insight into varieties that can be grown that are resistant to disease, using the elemental composition of the plant and perhaps by modulating of the soil characteristics to help slow the progression of this disease. The ionome characterization showed changes with the progression of the symptoms. In particular, the changes in Ca levels in the Leccino show that the previous observed response in tobacco, grapes, pecans, and blueberries is conserved in this pathosystem.

## 4. Materials and Methods

### 4.1. Field Survey Samples Collection

Leaf samples from naturally infected olive trees were collected in late November 2017 from two different olive groves located in the area declared *X. fastidiosa*-infected (Figure 1), in the south of the Apulia region (Italy): Field 1 (Gallipoli, Lecce, 40.0233750, 18.0516470), Field 2 (Sannicola, Lecce, 40.124325, 18.045239) (Figure 1). In each field, 30-year-old olive trees, grown under the same agronomic management, were selected for Leccino and Ogliarola salentina cultivars (Table 7). For each tree, five infected asymptomatic branches and five infected symptomatic branches were selected, and mature leaves were detached from the median part of hardwood cuttings (approx. 2 years old) and collected for analysis (Table 7). Uninfected olive trees could not be retrieved in the area selected for testing, despite the lack of symptoms.

### 4.2. DNA Extraction and Bacterial Quantification

Extraction of total DNA from olive samples was performed according to the cetyltrimethylammonium bromide (CTAB) protocol reported by Loconsole et al. (2014), using leaf petioles from the same infected asymptomatic and infected symptomatic tissues used for ionome determination. The presence of *X. fastidiosa* within the plant tissues was assessed by quantitative polymerase chain reaction (qPCR) using the protocol previously described by Harper et al. [69], with TaqMan^®^ Fast Advanced Master Mix (Thermo Fisher Scientific, Waltham, MA, USA) on a CXF 96™ Real-Time System (BioRad Laboratories, Hercules, CA, USA). Reactions were performed using 100 ng/μL total DNA. Ten biological replicates for each cultivar in Field 1 and four biological replicates for each cultivar in Field 2. A pool of ten leaf petioles representing ten portions of the plant, five infected symptomatic and five infected asymptomatic, was used for bacterial detection and quantification in each plant. Each sample was run in duplicate, and the data were averaged.

To estimate the bacterial population within the tissues, we established a standard curve using known concentrations of *X. fastidiosa* DNA. Ten-fold serial dilutions of an inactivated bacterial suspension with an initial OD_600_ of 0.5, corresponding to ca. 10^8^ CFU/mL, were prepared to obtain dilutions ranging from 10^7^ to 10^1^ CFU/mL. Each dilution was spiked in homogenized plant tissues of a not *Xf*–infected olive prior to DNA extraction. Bacterial concentration from Leccino and Ogliarola salentina samples was inferred by the standard calibration curve using Cqs from qPCR.

### 4.3. Ionome Characterization

Characterization of ionome of olive leaves was done as previously described [52], with some modifications. Briefly, leaves were prepared for analysis by drying at 65 °C overnight. Dried samples were then crushed to a fine powder by mortar and pestle and sampled at 10 mg of dry weight. Samples were digested for 1 h at 100 °C in 200 µL of mineral-free concentrated nitric acid (OPTIMA, Fisher Scientific International Inc., Pittsburgh, PA, USA). After dilution with ultra-pure, mineral-free water and centrifugation at 13,000× *g* to remove any particulates, samples were analyzed by inductively coupled plasma with optical emission spectroscopy (ICP-OES 7300 DV, PerkinElmer, Waltham, MA, USA) with simultaneous measurement of Ca, Cu, iron (Fe), potassium (K), Mg, manganese (Mn), sodium (Na), sulfur (S), phosphorus (P), and Zn. Minerals and trace element concentrations were determined by comparing emission intensities to a standard curve created by certified standards (CertiPrep, SPEX, Metuchen, NJ, USA).

To assess the ionome of the two varieties, an analysis of five samples randomly selected from the two fields were used to identify a mineral element that could be used as an internal denominator for data normalization. Variable dry weights (2.5, 5, 10, 15, and 20 mg) were analyzed and showed a linear response in total elements measured with correlation for individual samples of R^2^ > 0.95 (data not shown). When combining data from multiple varieties, R^2^ varied from 0.71 to 0.95 (data not shown). The higher the R^2^, the more the element remains unchanged in variety and field. On this basis, the most reliable elements for detection, with an R^2^ greater than 0.95, were P and K. Data were normalized based on these internal denominators (P and K concentration) and converted to dry weight based on the linear regression equation (for K y = 3.98x + 0.6972, for P y = 1.63x − 0.5249).

### 4.4. Soil Sampling and Physicochemical Analysis

Topsoil samples, 10–15 cm in depth, were randomly collected from four spots at each field used for our survey. Spots were chosen as pair replicates, sampled in the areas occupied by trees of each of the two olive varieties. To homogenize and remove plant debris and stones, the soil was first mixed and sieved through a 2 mm mesh. The determination of standard chemical and physical properties was conducted according to the accredited Italian soil study methods. Total nitrogen content was determined by mineralization and distillation procedure according to the Kjeldahl method [70]. The content of exchangeable Ca, Mg, K, and Na ions, removed from the exchange sites with buffered barium chloride (BaCl_2_) solution at pH 8.2, was determined by flame atomic absorption spectroscopy (FAAS, Shimadzu Japan). Soluble boron (B) was extracted by treating samples with a dilute aqueous solution of calcium chloride (CaCl_2_) (ratio 2:1). The content of the element was determined by spectrophotometry using the azomethine-H method. The active limestone content was determined by cold reaction with ammonium oxalate and titration with potassium permanganate (KMnO_4_).

Cation exchange capacity (CEC) was measured with a buffered BaCl_2_ solution at pH 8.2 and complexometric titration with triethanolamine. Organic carbon was determined by oxidation with potassium dichromate (K_2_Cr_2_O_7_), in the presence of sulphuric acid (H_2_SO_4_), and titration with iron (II) sulphate heptahydrate, according to Walkley–Black method. Assimilable metals, that is, Fe, Mn, Cu, and Zn, were extracted with a solution of diethylenetriaminepentaacetic acid/CaCl_2_/triethanolamine at pH 7.3, following the procedure described by Lindsay and Norwell [71]. Assimilable P content was determined by spectrophotometric FAAS measurement using ascorbic acid (Olsen method). Electrical conductivity and pH were determined from aqueous soil extracts (1:2.5 *w*/*v*).

### 4.5. Statistical Analysis

Statistical analysis was performed with the analysis toolpack of Microsoft Office Excel^®^ 2016 for Windows software (Microsoft Corporation, Redmond, WA, USA). Individual mineral and trace element concentrations in each variety and field were analyzed by pairwise comparisons using the two-tailed Student’s *t*-test. Statistical significance was accepted for *p*-values < 0.05. A comparison of the ionome of the cultivars in the same field was investigated to identify possible phenotypic markers that affect the susceptibility of the individual varieties, using principal component analysis. Heat maps, and hierarchical analysis were generated using CLUSTVIS web tool [72], using unit variance scaling and singular value decomposition (SVD) with imputation. A factorial univariate ANOVA analysis was performed to assess the effect of three independent variables (field, status, and variety) separately, as well as the joint effect of their interactions, with respect to three elements Ca, Mn, and Na. Statistical significance was accepted for *p*-values < 0.05.

## Figures and Tables

**Figure 1 pathogens-08-00272-f001:**
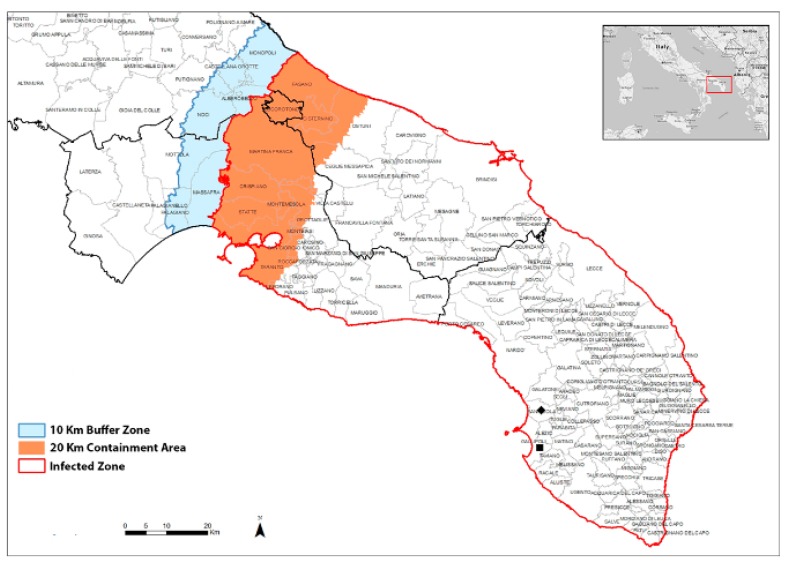
Map of the demarcated areas established in the region of Apulia (Italy), as defined by Commission Implementing Decision (EU) 2018/927 of 27 June, 2018, with indication of the experimental sites used in this study: ■ Field 1 (Gallipoli. Lecce. 40.0233750, 18.0516470); ◆ Field 2 (Sannicola. Lecce. 40.124325, 18.045239). The original map data were from Google Maps (DigitalGlobe, Westminster, CO, USA). This version was created using Adobe^®^ Photoshop CS6 (Adobe Systems Inc. San Jose, CA, USA) software.

**Figure 2 pathogens-08-00272-f002:**
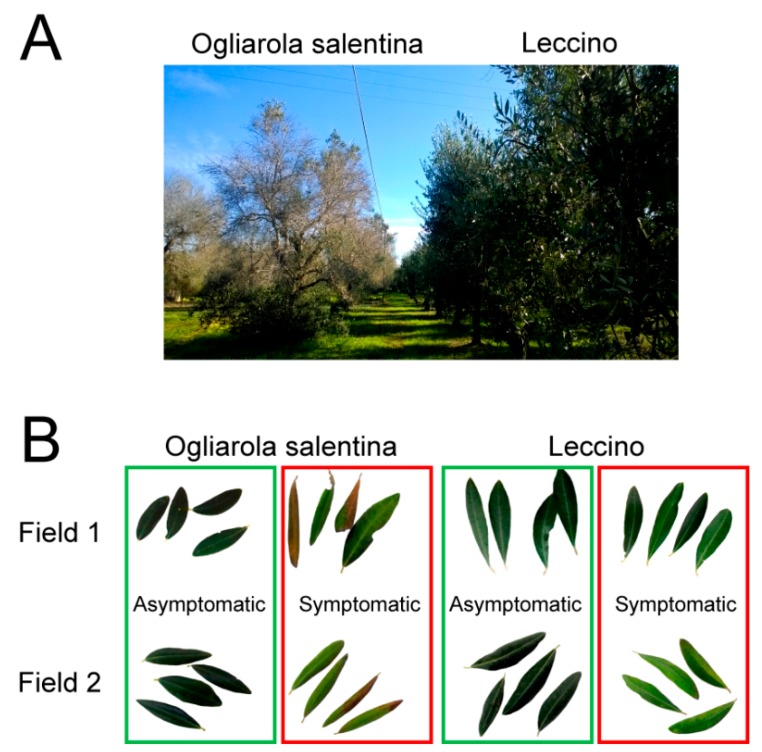
Evidence of different symptomatic response to *Xylella fastidiosa* infection. (**A**) Shows heavily susceptible olive Ogliarola salentina (left), in comparison with resistant Leccino (right), observed in a grove located in the olive quick decline syndrome-affected area (Alezio, Lecce, Italy). (**B**) A representative sample of the olive leaves used for ionome analysis. The image shows leaves chosen from infected symptomatic (boxed in red) and infected asymptomatic (boxed in green) branches of each of the two varieties, that is, Ogliarola salentina (left) and Leccino (right) and sampled in each of the two fields surveyed in this study, Field 1 and Field 2.

**Figure 3 pathogens-08-00272-f003:**
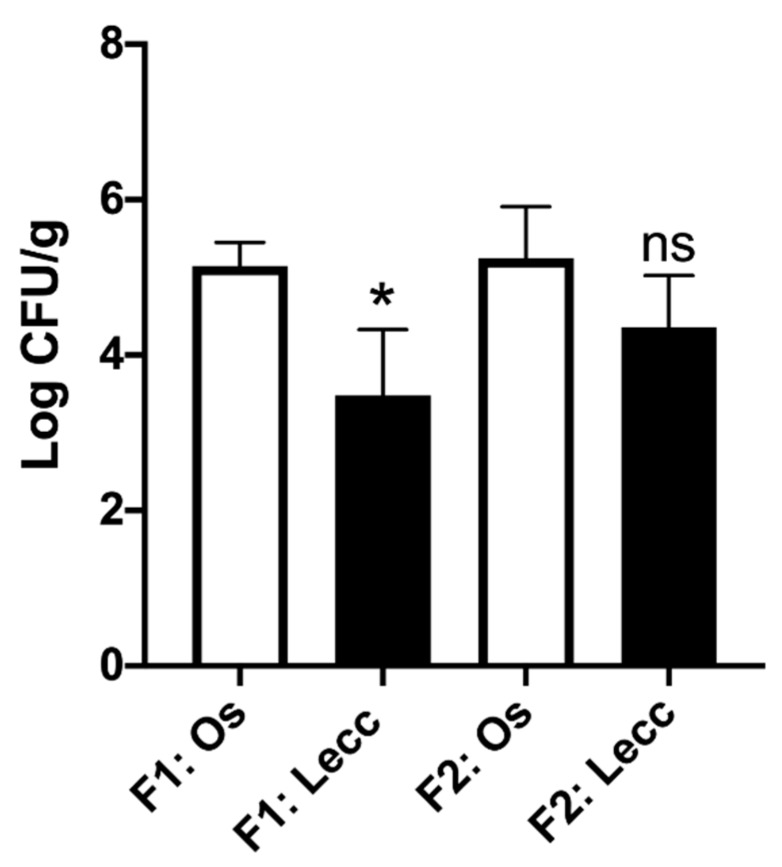
*Xylella fastidiosa* population in leaf samples of olive Ogliarola salentina and Leccino, quantified by qPCR assay. Bacterial concentration is expressed as log CFU/g, inferred using a standard calibration curve. Ten biological replicates for each cultivar in Field 1 (F1: Os for Ogliarola salentina and F1: Lecc for Leccino) and four biological replicates for each cultivar in Field 2 (F2: Os for Ogliarola salentina and F2: Lecc for Leccino). A pool of ten leaf petioles representing ten portions of the plant, five infected symptomatic and five infected asymptomatic, was used for bacterial detection and quantification in each plant. Each point represents the average of two independent experiments. There was a statistically significant difference in Field 1 (t-value = −5.54; df = 9; *p*-value = 0.0003 for α level 0.05), while no significant difference in Field 2 (t-value = −2.08; df = 3; *p*-value = 0.128 for α level 0.05), according to two-tailed Student’s *t*-test pairwise comparisons. Bars represent standard deviation of the mean, * indicates statistically significant differences (*p* < 0.05), and ns indicates no statistically significant difference (*p* > 0.05) between the two varieties.

**Figure 4 pathogens-08-00272-f004:**
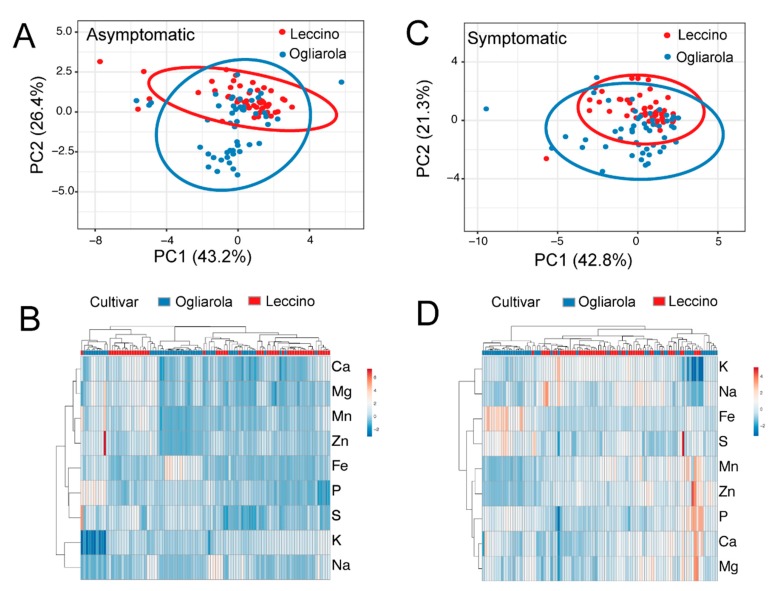
Principal component analysis (**A**,**C**) of leaves from Field 1, with prediction ellipses that show the probability of 0.95 that a new observation will fall in the same group. In the heat map representation of elemental composition, both rows and columns are clustered using correlation distance and average linkage (**B**,**D**). Infected asymptomatic leaves are shown in (**A**,**B**), and infected symptomatic leaves in (**C**,**D**). Each point represents an individual sample, as outlined in the materials and methods (*n* = 100 per cultivar).

**Figure 5 pathogens-08-00272-f005:**
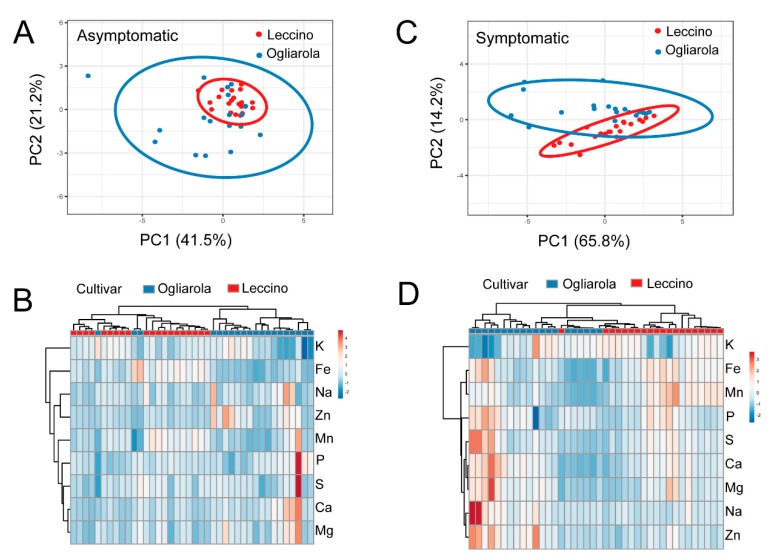
Principal component analysis (**A**,**C**) of leaves from Field 2, with prediction ellipses that show the probability of 0.95 that a new observation will fall in the same group. In the heat map representation of elemental composition, both rows and columns are clustered using correlation distance and average linkage (**B**,**D**). Infected asymptomatic leaves are shown in (**A**,**B**), and infected symptomatic leaves in (**C**,**D**). Each point represents an individual sample, as outlined in the materials and methods (*n* = 40 per cultivar).

**Table 1 pathogens-08-00272-t001:** Elemental analysis of olive leaves expressed in weight per dry weight.

		Ca	K	Mg	S	P	Fe	Mn	Na	Zn
Reference *	g/kg or mg/kg	10–14	8–10	1–1.6	0.8–1.6	1–1.3	90–124	20–36	<200	4–9
	**Field 1**									
Leccino	Asymp	13.7	3.9	1.8	1.6	1.3	41.5	43.4	33.0	9.3
	Symp	16.2	4.2	1.6	1.5	1.4	45.6	44.3	59.1	9.5
Ogliarola salentina	Asymp	13.3	3.1	1.6	1.9	1.5	96.6	30.4	25.5	6.7
	Symp	12.9	3.1	1.6	1.9	1.3	95.3	31.5	61.0	8.8
	**Field 2**									
Leccino	Asymp	6.4	4.2	1.0	1.5	1.2	63.4	44.4	15.3	7.1
	Symp	11.3	3.8	1.5	1.6	1.6	87.6	80.5	48.9	10.2
Ogliarola salentina	Asymp	10.6	3.8	1.5	1.7	1.2	42.1	29.6	23.1	10.7
	Symp	14.6	3.4	2.0	1.8	1.7	60.6	40.3	126.8	18.4

Field 1 (Gallipoli, Lecce, 40.0233750, 18.0516470); Field 2 (Sannicola, Lecce, 40.124325, 18.045239). Element concentrations are expressed in g/kg, except for Fe, Mn, Na, and Zn that are expressed in mg/kg. * Reference concentrations were obtained from Kailis and Harris [34].

**Table 2 pathogens-08-00272-t002:** Physico-chemical analysis of soils wherein the leaf ionome profiles were evaluated.

		Field 1		Field 2	
	Reference Range	Ogliarola Mean ± St. Dev	Leccino Mean ± St. Dev	Ogliarola Mean ± St. Dev	Leccino Mean ± St. Dev
**Texture**					
Skeleton (>2 mm) %	0–10	0	0	1 ± 1.4	0 ± 0.00
Sand (2–0.05 mm) % Fine Fraction	-	51 ± 3	51 ± 3	47 ± 6	37 ± 3
Silt (0.05–0.002 mm) % Fine Fraction	-	25 ± 1	24 ± 4	32 ± 6	34 ± 0
Clay (<0.002 mm) % Fine Fraction	-	24 ± 4	25 ± 1	21 ± 0.7	29 ± 3
**Chemical and Physical Parameters**					
pH in H_2_O (1:2.5) mS∙cm^2^	6.7–7.3	7.05 ± 0.49	8.05 ± 0.07	8.1 ± 0.14	7.3 ± 0.28
**Macro Elements**					
Assimilable phosphorus (mg∙kg^−1^ Fine Fraction)	27–32	70 ± 10	34 ± 3	7 ± 1	5 ± 0.9
Exchangeable potassium (mg∙kg^−1^ Fine Fraction)	200	310 ± 104	232 ± 52	355 ± 76	102 ± 8
Exchangeable calcium (mg∙kg^−1^ Fine Fraction)	3000	1601 ± 126	2231 ± 344	2610 ± 25	2130 ± 632
Exchangeable sodium (mg∙kg^−1^ Fine Fraction)	230	63 ± 27	96 ± 11	54 ± 1	37 ± 4
Exchangeable magnesium (mg∙kg^−1^ Fine Fraction)	200	211 ± 54	284 ± 104	322 ± 42	427 ± 88
**Micro Elements**					
Assimilable iron (mg∙kg^−1^ Fine Fraction)	5–40	16 ± 8	6 ± 0.3	4.4 ± 0.3	7.2 ± 1.3
Assimilable manganese (mg∙kg^−1^ Fine Fraction)	5–20	38 ± 7	27 ± 5	27 ± 3	57 ± 7.4
Assimilable copper (mg∙kg^−1^ Fine Fraction)	1–5	30 ± 2	22 ± 0.2	11.3 ± 1.7	8.6 ± 0.1
Assimilable zinc (mg∙kg^−1^ Fine Fraction)	0.5–2	1 ± 0.3	1.0 ± 0.01	1.5 ± 0.1	0.6 ± 0.04

Values represent averages of two replicate samplings. Ogliarola or Leccino refers to soil collected from the same area where Ogliarola salentina or Leccino trees used in this study were found, respectively. Field 1 (Gallipoli, Lecce, 40.0233750, 18.0516470); Field 2 (Sannicola, Lecce, 40.124325, 18.045239).

**Table 3 pathogens-08-00272-t003:** Variance explained by principal components in PCA of element concentration of infected asymptomatic and infected symptomatic leaves of each olive variety in different fields.

		PC1	PC2	PC3	PC4	PC5	PC6	PC7	PC8
**Field 1**	Asymp	0.43	0.26	0.10	0.06	0.05	0.04	0.03	0.02
	Symp	0.43	0.21	0.13	0.10	0.06	0.03	0.03	0.01
**Field 2**	Asymp	0.42	0.21	0.15	0.09	0.07	0.03	0.02	0.01
	Symp	0.66	0.14	0.11	0.05	0.01	0.01	0.01	0.00

**Table 4 pathogens-08-00272-t004:** Component loadings of principal components 1 and 2 from the PCA of element concentration in infected symptomatic leaves versus infected asymptomatic leaves of each olive variety in different fields.

			**CA**	**FE**	**K**	**MG**	**MN**	**NA**	**P**	**S**	**ZN**
**Field 1**	Asymp	PC1	0.40	0.17	−0.46	0.38	0.22	−0.27	0.46	0.27	0.22
		PC2	0.16	−0.52	0.12	0.24	0.47	0.29	−0.12	−0.31	0.47
	Symp	PC1	−0.37	−0.14	0.37	−0.41	−0.39	0.04	−0.37	−0.32	−0.38
		PC2	0.33	−0.45	0.39	0.23	0.31	0.37	−0.40	−0.18	0.23
**Field 2**	Asymp	PC1	0.49	0.07	−0.16	0.46	0.29	0.21	0.38	0.42	0.26
		PC2	−0.07	0.44	−0.30	−0.16	0.20	−0.44	0.35	0.17	−0.55
	Symp	PC1	0.38	0.33	−0.33	0.38	0.26	0.28	0.32	0.39	0.32
		PC2	0.04	−0.35	0.19	0.03	−0.49	0.55	−0.21	0.10	0.49

**Table 5 pathogens-08-00272-t005:** Percent difference in element concentration in leaves of olive Leccino versus Ogliarola salentina, separated by visual symptomatology.

		Ca	K	Mg	S	P	Fe	Mn	Na	Zn
**Location** ^#^	**Infected Asymptomatic**									
Field 1	% difference *	4	**23**	14	−11	**−15**	**−57**	**43**	**30**	**39**
	*p*-value	0.53	6 × 10^−4^	0.08	0.14	6 × 10^−4^	1 × 10^−7^	7 × 10^−5^	2 × 10^−2^	6 × 10^−3^
Field 2	% difference *	**−40**	11	**−34**	−5	−10	**51**	**50**	−34	**−34**
	*p*-value	6 × 10^−3^	0.10	2 × 10^−3^	0.59	0.11	6 × 10^−3^	2 × 10^−3^	0.14	4 × 10^−3^
	**Infected Symptomatic**									
Field 1	% difference *	**21**	**35**	−1	4	**−23**	**−52**	**41**	−3	8
	*p*-value	2 × 10^−2^	9 × 10^−7^	0.93	0.59	8 × 10^−7^	3 × 10^−8^	2 × 10^−3^	0.85	0.56
Field 2	% difference *	−22	13	−25	−7	−10	45	**100**	**−61**	**−45**
	*p*-value	0.24	0.25	0.17	0.68	0.24	0.043	2 × 10^−4^	1 × 10^−2^	2 × 10^−3^

^#^ Field 1 (Gallipoli, Lecce, 40.0233750, 18.0516470); Field 2 (Sannicola, Lecce, 40.124325, 18.045239). * Numbers indicate the percent of difference in element concentration in the leaves of Leccino compared with Ogliarola salentina olive cultivars. For each comparison, either infected asymptomatic or infected symptomatic leaves were used. Numbers in bold indicate significant differences according to Student’s *t*-test (*p*-values < 0.05).

**Table 6 pathogens-08-00272-t006:** Percent difference in element concentration in infected symptomatic leaves versus infected asymptomatic leaves of each olive variety.

		Ca	K	Mg	S	P	Fe	Mn	Na	Zn
	**Field 1**									
Leccino	% difference *	**19**	9	−12	−9	3	10	2	**79**	2
	*p*-value	2 × 10^−2^	6 × 10^−2^	1 × 10^−1^	7 × 10^−1^	5 × 10^−2^	2 × 10^−1^	8 × 10^−1^	1 × 10^−3^	7 × 10^−1^
Ogliarola salentina	% difference *	−3	−1	2	0	−12	−1	4	**139**	31
	*p*-value	7 × 10^−1^	9 × 10^−1^	8 × 10^−1^	7 × 10^−2^	9 × 10^−1^	9 × 10^−1^	7 × 10^−1^	1 × 10^−5^	1 × 10^−1^
	**Field 2**									
Leccino	% difference *	**76**	−8	**56**	**8**	33	**38**	**81**	**220**	**45**
	*p*-value	6 × 10^−4^	1 × 10^−1^	1 × 10^−2^	6 × 10^−3^	1 × 10^−1^	7 × 10^−3^	1 × 10^−4^	7 × 10^−5^	5 × 10^−3^
Ogliarola salentina	% difference *	37	−10	36	8	36	44	36	**449**	**72**
	*p*-value	1 × 10^−1^	6 × 10^−1^	9 × 10^−2^	1 × 10^−1^	7 × 10^−1^	1 × 10^−1^	7 × 10^−2^	5 × 10^−4^	3 × 10^−3^

* Numbers indicate the percent of difference in element concentration in infected symptomatic vs. infected asymptomatic leaves of either Leccino or Ogliarola salentina olive cultivars. Comparisons were made separated for each field. Field 1 (Gallipoli, Lecce, 40.0233750, 18.0516470); Field 2 (Sannicola, Lecce, 40.124325, 18.045239). Numbers in bold indicate significant differences according to Student’s *t*-test (*p*-values < 0.05).

**Table 7 pathogens-08-00272-t007:** Summary of olive samples analyzed by ICP-OES and qPCR.

Field	Variety	Trees	Replicate Samples	Total Samples
Field 1	Leccino	10	5 inf. symptomatic–5 inf. asymptomatic	100
	Ogliarola salentina	10	5 inf. symptomatic–5 inf. asymptomatic	100
Field 2	Leccino	4	5 inf. symptomatic–5 inf. asymptomatic	40
	Ogliarola salentina	4	5 inf. symptomatic–5 inf. asymptomatic	40
**Total**		**28**		**280**

Field 1 (Gallipoli, Lecce, 40.0233750, 18.0516470); Field 2 (Sannicola, Lecce, 40.124325, 18.045239).

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
