# Peer review of "Ionomic Differences between Susceptible and Resistant Olive Cultivars Infected by Xylella fastidiosa in the Outbreak Area of Salento, Italy"

_pathogens, 2019, doi:10.3390/pathogens8040272_

Round 1

Reviewer 1 Report

This is a well written paper that presents a descriptive study on the association of leaf mineral ions with resistance and susceptibility of two different olive tree varieties to Xylella fastidiosa.  This is a very impactful emerging disease and this study usefully contributes to the knowledge base on this issue.  I think the study was well conducted and I only have relatively minor suggestions for improvement in the attached annotated manuscript.  Very importantly, however, I caution the authors to avoid assigning causation to association, as they seem to do in some cases (e.g. l. 327-328).

I think the figures and tables are all necessary and useful for the understanding of the story.  However, I think the number of references is excessive and could be reduced somewhat (perhaps 1/3?) without sacrificing support.

Author Response

Reviewer 1

We would like to thank the editor and the reviewer for their thorough review. We have made all the corrections and additions to the text as suggested. We believe that these changes as suggested by the reviewers have enhanced the readability and completeness of the article, and we are very grateful for this.

Rev 1. importantly, however, I caution the authors to avoid assigning causation to association, as they seem to do in some cases (e.g. l. 327-328).

Author Response: We thank the reviewer for highlighting this in the text. We have changed all cases of “changes” for “difference” which is what we are truly measuring. We agree with the reviewer that claims of causation will require further experiments to we have amended all those statements to make it clear.

Rev 1: the number of references is excessive and could be reduced somewhat (perhaps 1/3?) without sacrificing support.

Author Response: We have reduced the number of citations in the text by 16 of 88 total originally.

From the annotation:

We thank the reviewer for the changes highlighted in the text and we have made all the suggestions (within the framework of other reviewer suggestions). We would like to highlight two significant changes suggested by the reviewer.

Rev1: Throughout the paper be very careful with the concepts of tolerance and resistance, which are quite different. I would use only resistance, which is a gradient of responses, as opposed to immunity, which is a yes/no trait.  Tolerance has a completely different meaning.  See APS glossary: tolerance - the ability of a plant to endure an infectious or noninfectious disease, adverse conditions, or chemical injury without serious damage or yield loss; resistant - possessing properties that prevent or impede disease development (contrasts with susceptible)

Author response: We have changed all cases to resistance when referring to ‘Leccino’ variety in the manuscript. Field and greenhouse testing show the lower concentration of the bacterium in this cultivar compared to ‘Ogliarola salentina’, therefore supporting the definition.

Rev1: (paraphrased) Please clarify the role ROS and manganese in possible protection.

Author response: In the discussion regarding increased manganese content we have clarified the text to make it clear that balancing of Mn is important: We have edited the section to read as follows.  

On this basis, elevated Mn in ‘Leccino’ could allow for enhanced lignin and flavonoids production to help protect the host. Simpson et al. [66] have suggested that the uptake of Fe and other transition metal ions such as Mn, by X. fastidiosa, cause the exhaustion of essential micronutrients in xylem, leading, in citrus trees, to symptoms typical of variegated chlorosis. Our data could suggest the elevated level of Mn in ‘Leccino’ could more effectively counterbalance any deficit triggered by the bacterial infection. Other examples are Mn dependent enzyme are: superoxide dismutase [54], oxalate oxidase [55] and the oxygen-evolving complex of PSII [56-58]. Production of ROS, termed “oxidative burst”, is a hallmark of successful recognition of infection and activation of plant defense [59], since it can directly act as antimicrobial agent during the plant defense response [60-63] however, it also can cause damage if not controlled. Therefore, increased Mn for superoxide dismutase may help to balance the benefit and cost of ROS production.

Rev1: (paraphrased) Please improve the description of the PCA.

Author response: We have edited the description of the principal component analysis to use more precise language as follows.

Principal component analysis suggests that ‘Leccino’ and ‘Ogliarola salentina’ have differences in their mineral and trace element content (Figure 4A, 4C and Figure 5A, 5C). The variance explained by each principal component is shown in Table 3 and the component loadings for PC1 and PC2 are shown in Table 4. The PCA component values were used for hierarchical cluster analysis and generation of a heat map (Figure 4B, 4D and Figure 5B, 5D). In general, the varieties were separated into independent clades with some overlap in ionome characteristics. We did not determine the variables contributing to this overlap/clustering together.

Reviewer 2 Report

To the authors:

In the present study, titled as ‘Ionomic differences between susceptible and resistant olive cultivars infected by Xylella fastidiosa in the outbreak area of Salento, Italy’, the authors examined the mineral and trace elements content (ionome) of leaves from X. fastidiosa infected trees comparing two varieties (susceptible and tolerant), to show the possible cause of tolerance in Leccino trees against X. fastidiosa infection. In summary, the work is well structured, well-written and the results are worth it to be published.

As a reviewer, I appreciate the time and effort that was performed into the preparation of your article, only a minor revision is needed to address the concerns/questions mentioned in the manuscript (pdf). I recommend its publication after a minor revision.

Author Response

Reviewer 2

We would like to thank the editor and the reviewer for their thorough review. We have made all the corrections and additions to the text as suggested. We believe that these changes as suggested by the reviewers have enhanced the readability and completeness of the article, and we are very grateful for this.

From the annotations:

We thank the reviewer for the changes highlighted in the annotated text and we have made all the suggestions (within the framework of other reviewer suggestions). We would like to highlight significant changes suggested by the reviewer.

Rev2: Requested that all instances of “mineral elements” be changed to “mineral and trace elements”.

Author response: We have amended mineral to mineral and trace elements throughout the text

Rev2: According to general rule for statistics standard error should be use when the replicate number is very high (20). Is this the case here? If not, please put standard deviation (SD).

Author response: The total numbers of samples analyzed was less than 20 so we have remade Figure 3 with standard deviation rather the standard error of the mean.

Rev2: What is the reason for different numbers of trees?

Author Response: We have different numbers of trees due to the fact these are privately owned fields and it was difficult to find the trees in close proximity to each other of the stated varieties. 

Rev2: Suggests moving criteria for relevant difference to material and methods section

Author Response: This section describes the criteria for significant differences in mineral and trace elements. We feel as though it is I critical to include here in the discussion as in reinforces why some elements are discussed over others.

Rev 2 Suggests to move “this sentence” into the introduction section since it is an hypothesis.

Author response: We moved the general hypothesis to the introduction and reworded the second sentence that referred to a hypothesis so it was a discussion point for future work.

Rev2: Suggests that all ions are expressed with oxidation state.

Author response: We have chosen to express all elements without oxidation state as the methods we used does not discriminate between different states therefore we have left them in the generic form.

Reviewer 3 Report

The authors studied the “Ionomic differences between susceptible and resistant olive cultivars infected by Xylella fastidiosa in the outbreak area of Salento, Italy”. While I have no objection against publishing the data, I have some issues that need addressing. I am concerned about the poorly elements in the material and methods section; authors can provide details for results interpretation. The experimental set up of this study appears to be well-designed and the data collected carefully. However, information about some statistical values obtained in the experiments must be provided and detailed in the results section. I think that this manuscript requires substantial rewriting to make its results clearer and more readily interpretable to the reader. My specific comments are listed in the "Manuscript". Based on the comments above reported, my opinion is that this manuscript may be suitable for printing on this journal after corrections.

Author Response

Reviewer 3

We would like to thank the editor and the reviewer for their thorough review. We have made all the corrections and additions to the text as suggested. We believe that these changes as suggested by the reviewers have enhanced the readability and completeness of the article, and we are very grateful for this.

Rev 3: However, information about some statistical values obtained in the experiments must be provided and detailed in the results section.

Author response: We have added a description and all the statistical details requested by the reviewer for all experiments in the manuscript.

From the annotation

We thank the reviewer for the changes highlighted in the annotated text and we have made all the suggestions (within the framework of other reviewer suggestions). We would like to highlight significant changes suggested by the reviewer.

Rev3: Explain why they used these statistical methods. Please detail for each experiment.

Author response: We added the details to the figure legends, results section and materials and methods for each statistical test.

Round 2

Reviewer 3 Report

The manuscript “Ionomic differences between susceptible and resistant olive cultivars infected by Xylella fastidiosa in the outbreak area of Salento, Italy” has been improved and all my questions were taken into account.
I recommend the publication in “Pathogens”.